# Iodine status, household salt iodine content, knowledge and practice assessment among pregnant women in Butajira, South Central Ethiopia

**Senait Tadesse[1], Ariaya Hymete[1], Marya Lieberman[2], Seifu Hagos Gebreyesus[3], Ayenew Ashenef[1] ***

**1** Department of Pharmaceutical Chemistry and Pharmacognosy, School of Pharmacy, College of Health Sciences, Addis Ababa University, Addis Ababa, Ethiopia, **2** Department of Chemistry & Biochemistry, University of Notre Dame, Notre Dame, IN, United States of America, **3** Department of Nutrition and Dietetics, School of Public Health, College of Health Sciences, Addis Ababa University, Addis Ababa, Ethiopia

* ayenew.ashenef@aau.edu.et

## Abstract

### Background

Iodine is one of the crucial micronutrients needed by the human body, and is vitally important during pregnancy. This study aimed to determine the relationship between the iodine status of pregnant women and their knowledge, and practices regarding iodized salt. All participants were enrolled in the Butajira nutrition, mental health and pregnancy (BUNMAP) cohort, Butajira, Ethiopia in February-May, 2019.

### Methods

In this cross-sectional study, 152 pregnant women without hypertension or known thyroid disease before or during pregnancy were randomly selected from the BUNMAP mother to child cohort (n = 832). Spot urine samples were collected to estimate the level of urinary iodine concentration (UIC). Salt samples were also collected from their homes. The Sandall-Kolthoff (S-K) method was used to measure the level of iodine in the urine samples, and iodometric titration was used to measure the level of iodine in the salt. Data was entered and cleaned using Epi-info version 3.5.3 and then exported to SPSS version 20 for further analysis. Multivariate logistic regression analysis was performed to identify associations in the collected data.

### Results

The WHO recommended level of iodine for populations of pregnant women is 150–249 μg/L. The median UIC among pregnant women in this study was 151.2 μg/L [interquartile range (IQR) = 85.5–236.2 μg/L], at the low end of this range. About half (49.65%) of the participants were likely to be iodine deficient. There was a significant association between having a formal job (AOR = 2.56; CI = 1.11–5.96) and iodine sufficiency. Based on

**Data Availability Statement:** All data had been deposited in the open science frame work (OSF) https://osf.io/nrq5w/.

**Funding:** The authors thanks Addis Ababa University (AAU) graduate student research support programme. AAU thematic research project grant supported us for the Cohort study. The Ministry of Innovation and Technology, Government of Ethiopia National Innovation Award to AA helped us in some laboratory works.

**Competing interests:** NO authors have competing interests.

**Abbreviations:** BUNMAP, Butajira nutrition mental health and pregnancy; CCIDD, Council for control of iodine deficiency disorders; DSS, Demographic survey system; EDHS, Ethiopian health and demographic survey; EFDA, Ethiopian food and drug administration authority; EQUIP, Ensuring the quality of Urinary iodine procedures; ID, Iodine deficiency; IDD, Iodine deficiency disease; mUIC, Mean Urinary Iodine concentration; ppb, parts per billion; ppm, parts per million; UIC, Urinary Iodine concentration; UIE, Urinary iodine excretion.

a cutoff of >15 ppm (mg/kg), 91.7% (95% CI: 87.2–96.2) of the salts collected from the household had adequate iodine content. The median iodine level of the collected salt samples was 34.9 mg/kg (ppm) (IQR = 24.2–44.6 mg/kg).

## Conclusions

The UNICEF 2018 guidelines for adequate iodine nutrition in pregnant women include both a recommended median range of 150–249 μg/L, and an upper limit of 20% on the fraction of the population with UIC below 50 μg/L. Because our study population's median level is 151.2 μg/L and the percentage of pregnant women with urinary iodine concentration of less than 50 μg/L is 9.7% (14/145), the women received adequate iodine nutrition. The availability of adequately iodized salt in households is more than 90%, as recommended by WHO. In light of previous iodine deficiency in this region of Ethiopia, the salt iodization program promotes the health of babies and mothers.

## Introduction

Iodine is one of the crucial micronutrients required for normal growth of humans from the fetal stage to adulthood. Iodine is required for the synthesis of thyroxin ($T_3$) & triiodothyronine ($T_4$) hormones in the thyroid gland. These hormones regulate metabolism and protein synthesis as well as neuronal function and brain development [1]. When iodine intake is below recommended levels, the thyroid cannot synthesize enough thyroid hormones, resulting in many adverse effects on both individual and societal levels [2]. Adult iodine deficiency may lead to hypothyroidism and goiter [3]. Deficiency of iodine during pregnancy can cause stillbirth or neural defects in the child, and may increase the risk of spontaneous abortion [4]. Iodine deficiency during infancy causes permanent mental impairment. Based on the proportion of household consumption of adequately iodized salt, the United Nations Children's Fund (UNICEF) estimated that every year about 38 million newborns in developing countries are subjected to brain damage associated with iodine deficiency disorders (IDD) [5, 6]. WHO defined iodine deficiency as "the single most important preventable cause of brain damage in the world". People living in areas of iodine deficiency may have a lower intelligence quotient (IQ) than those in a comparable area with no iodine deficiency. This mental deficiency has an immediate effect on child learning capacity, quality of life in communities, and economic productivity [2].

Around 2 billion people have insufficient iodine nutrition worldwide [7]. Universal Salt iodization (USI) has been recommended to address this problem. By 2020, 124 countries had mandatory legislation and 21 countries had voluntary legislation requiring salt iodization. This has ensured that 81% of the world population has access to salt with sufficient iodine, with measurable impact on human health. From 1993 to 2019, the global prevalence of clinical IDDs (as assessed by the total goiter rate (TGR)) fell from 13.1% to 3.2%. 720 million cases of clinical IDDs have been prevented by USI (a reduction of 75.9%). USI has also reduced the number of newborns affected by IDDs, with 20.5 million cases prevented annually. Nevertheless 21 countries still have large numbers of people who do not receive sufficient iodine [8–10]. Dietary iodine requirements are higher in pregnant women than in non-pregnant adults because of increased thyroid hormone production, increased renal iodine losses and fetal iodine requirements in pregnancy [11]. WHO defines urinary iodine concentrations in the

population of pregnant women of <150 μg/L, 150–249 μg/L, 250–499 μg/L, and ≥500 μg/L as indicating insufficient, adequate, more than adequate, and excessive levels of iodine intake respectively [2].

In 2011, Ethiopia was the first among top ten iodine-deficient countries based on a national median UIC (20–49 μg/L) [12] among 148 countries data the study included in the analysis. The national goiter prevalence rate among children aged 6 to 12 years was 39.9%, representing more than 4 million children [13]. 66 million persons were estimated to be vulnerable to iodine deficiency as only 15% of households had access to iodized salt. Furthermore, 50,000 stillbirths from iodine-deficient women were observed annually and the goiter rate among the populations in communities across the country was 14–59% [13]. In 2014, Ethiopia was still one of the 25 most iodine-deficient countries, and only 19.9% of the population was reported to consume iodized household salt [14].

In Ethiopia, the food supply is quantitatively insufficient and not a balanced diet for many people, and includes few naturally iodine-rich foods. The Ethiopian government began to implement a salt iodization program in 2011, but research in different regions of Ethiopia show that the iodine deficiency problem still exists [15]. It is, therefore, necessary to continue to study the iodine status of pregnant women and to monitor the quality of iodized salt they use in order to take corrective action against iodine deficiency in this vulnerable group.

## Methods

### Study participants

A community-based cross-sectional study was conducted in one urban and six rural small administration units in Ethiopia. Each of the administration units is composed of localities/districts/collection of villages (locally known as *kebeles*) of Butajira. The study was conducted between February 10, 2019 and May 20, 2019. The study population consisted of pregnant women without thyroid disease or hypertension, who lived in the selected rural and urban areas. 152 women were selected randomly from among 832 pregnant women enrolled in the BUNMAP mother to child cohort and asked to participate in this study; nearly all agreed, giving a sample size of 145 women.

Study participants consented in written form after the possible risks, benefits, and purposes of the project were explained clearly to them. All possible issues of confidentiality were addressed. Their participation was based on pure voluntarism with clear understanding that they could withdraw from the study at any time. The investigation is carried out in compliance with the principles of the declaration of Helsinki. Data were stored in password-protected computers and hard copies were kept in locked files at the project office in Addis Ababa, Ethiopia. In accordance with the protocol, pregnant women who agreed to be included in the cohort were offered transportation reimbursements for prenatal care attendance in the nearby health centers and hospital. Free care (such as ultrasound imaging check-ups every three months) and prenatal care were given by gynecology specialists who are members of the faculty at Addis Ababa University, School of Medicine that were availed by the project. Approval for the study was obtained from the Ethical Review committee of the School of Pharmacy, Addis Ababa University (AAU) dated November 30, 2018 referenced as ERB/SOP/43/11/2018 and IRB of the College of Health Sciences, AAU. Purpose of the study was explained to all study participants, and written informed consent obtained. The study was carried out in accordance with relevant guidelines and regulations.

The sample size was calculated using a single population proportion formula (see below) based on the following assumption, expected prevalence of iodine deficiency 90% [16], absolute sampling error of five, and a 95% confidence interval [17]. Hence, the sample size was 152

with a10% non-response rate. The number of pregnant women enrolled in the cohort at the time of this data collection was 832. Hence, we applied a simple random sampling method to select women from enrolled pregnant women.

$$N = \frac{Z^2 \times P(1-P)}{d^2} = \frac{1.96^2 \times 0.9(1-0.9)}{0.5^2} = 138$$

Where

N = Sample size

P = Prevalence of iodine deficiency in pregnant women [16]

Z = Z-score

d = Sampling error

The BUNMAP is a population based cohort established in 2016. It consists of pregnant women and their offspring living in selected clusters of the Butajira Health and Demographic Surveillance Site (BHDSS), South Central Ethiopia. This study site is one of the oldest surveillance sites in Africa, established in 1986. It is managed by Addis Ababa University, School of Public Health and serves as community training and research center for the College of Health Sciences at Addis Ababa University. A semi-structured interviewer-administered questionnaire was used to collect data [as shown as S1 File]. The questionnaire was first prepared in English, then translated into the local language (Amharic) and finally, back-translated to English to maintain consistency. Quality assurance mechanisms included training the data collectors and supervisors about the objectives of the project and the data collection process. Pretesting of the questionnaire was carried out to check suitability of the tool to the study objective. Approximately 30–50 grams of a salt sample was collected from the participants' homes and kept in air-tight plastic containers. The samples were stored at room temperature, away from sunlight, until analysis was carried out. About 10 ml– 20 ml of spot urine samples were collected from the participants in labeled and sterilized screw-capped plastic bottles. The urine samples were stored on ice during transportation to Addis Ababa, and were then kept at 4˚C in a refrigerator in tightly closed containers until analysis. The salt iodine content was estimated in ppm by an iodometric titration method [2]. The urinary iodine level was estimated in ppb by the Sandall-Kolthoff (S-K) spectrophotometric method [2]. Quality control samples (urine samples in which the iodine content (UIC) was determined at a certified laboratory) were obtained from the USA for the laboratory quality assurance purposes. Data were entered into Epi-info version 7.2.2.6 (CDC, Atlanta, USA) and analyzed using Statistical Package for Social Sciences (SPSS 20.0) (IBM Corp., Armonk, NY, USA). Descriptive statistics were used for frequency distribution. The strengths of the associations between the dependent variable (UIC) and independent variables (various factors reported by the participants) were assessed via multiple regressions to produce adjusted odds ratio (AOR) values and 95% confidence intervals. When the 95% CI for the AOR value for a factor did not include 1, the impact of that factor was judged to be significant.

## Results

We asked 152 women selected randomly from pregnant women in the BUNMAP mother to child cohort to participate in this study, and 145 agreed. The sociodemographic characteristics and obstetric history of the study participants are shown in Table 1.

A total of 152 pregnant women were randomly selected for the study. The response rate was 95.4% (n = 145). The mean age of the pregnant women was 26 ± 4.6 years and the

**Table 1. Characteristics of pregnant women (N = 145) in BUNMAP cohort, Butajira, Ethiopia, February, 2019 (at the time of recruitment).**

| Variable | Frequency | Percentage |
|---|---|---|
| Age group | | |
| 18–24 | 50 | 34.5 |
| 25–34 | 85 | 58.6 |
| 35–49 | 10 | 6.9 |
| Educational status | | |
| Not able to read and write | 33 | 22.8 |
| Able to read and write but no formal education | 7 | 4.8 |
| Primary education | 83 | 57.2 |
| Secondary education | 19 | 13.1 |
| Tertiary education | 3 | 2.1 |
| Marital status | | |
| Married | 145 | 100 |
| Single | - | 0 |
| Occupation | | |
| Merchant | 48 | 33.1 |
| Government employee | 3 | 2.1 |
| Housewife | 94 | 64.8 |
| Monthly family income (in Ethiopian birr*) | | |
| ≤ 500 | 46 | 31.7 |
| 501–1000 | 41 | 28.3 |
| 1001–2500 | 51 | 35.2 |
| 2501–5000 | 7 | 4.8 |
| Family size | | |
| <5 persons | 104 | 71.7 |
| ≥ 5 persons | 41 | 28.3 |
| Gestational trimester | | |
| First | 23 | 15.9 |
| Second | 47 | 32.4 |
| Third | 75 | 51.7 |
| Number of children | | |
| 0 | 48 | 33.1 |
| 1 | 40 | 27.6 |
| 2–4 | 46 | 31.7 |
| ≥ 5 | 11 | 7.6 |
| Residence | | |
| Rural | 58 | 40 |
| Urban | 87 | 60 |

[*USD equivalent conversion factor as of Nov. 8, 2021, was 100 Birr = $2.11].

majorities (93%) were below 34 years. Approximately half (57.2%) of the women had attended primary level education (up to grade six that may take six academic years of schooling in formal elementary schools) while 2.1% had tertiary level education (graduating from a university in any field that may took 3–6 academic years depending on the subject of study). More than half (64.8%) of the women listed their occupation as housewives. The mean family's monthly income was 1135.2±759.62 Ethiopian birr (which is equivalent to $24±16 USD). About one-

third (35.2%) of the respondents had a family's monthly income between 1001 to 2500 Ethiopian birr ($21–53 USD). The mean number of children in a family for the woman was 1.55 ± 1.62. The mean family size was 3.37±1.7 persons. Almost half (51.7%) of the women included in the study were in their third trimester and about one-third (33.1%) of the women were primiparous.

## Iodine content of household salt

The household salt iodine levels found in this study are depicted in Fig 1.

WHO recommends that the iodine level of household salt should be in the range of 15 ppm– 40 ppm for the successful prevention of IDDs. The average iodine level of household salt in this area was in the recommended range, and only 12 out of 145 samples were below the recommended range. The median iodine level of the collected salt samples was 34.9 ppm (IQR = 24.2–44.6).

In this study, a major proportion (91.7%, 95%$CI$ = 87.2% - 96.2%) of households were consuming adequately iodized salt ($\geq$ 15 ppm). Of these households, 62 (42.7%) provided salt samples that contained more than 40 ppm iodine. Only 12 (8.3%) of the household salt samples contained less than 15 ppm (Fig 1).

## Knowledge and practice of pregnant women about iodized salt

More than half of the women (55.2%) consumed coarse (non-packaged to mean as individual units for sale) salt while 65 (44.82%) used packaged (powdered) salt. About 58.6% of women claimed that they are using iodized salt while 41.4% claimed that they have non-iodized salt for daily consumption. Of the 85 women who reported that they thought they were using iodized salt, 77 (90.6%) actually were using salt that was adequately iodized. In contrast, of the 60 women who believed that they did not use iodized salt, 56 (93.3%) were actually using salt that was adequately iodized.

The knowledge of the women about iodized salt was assessed by asking four questions about iodized salt and its health benefits; if a woman answered two of them correctly she was

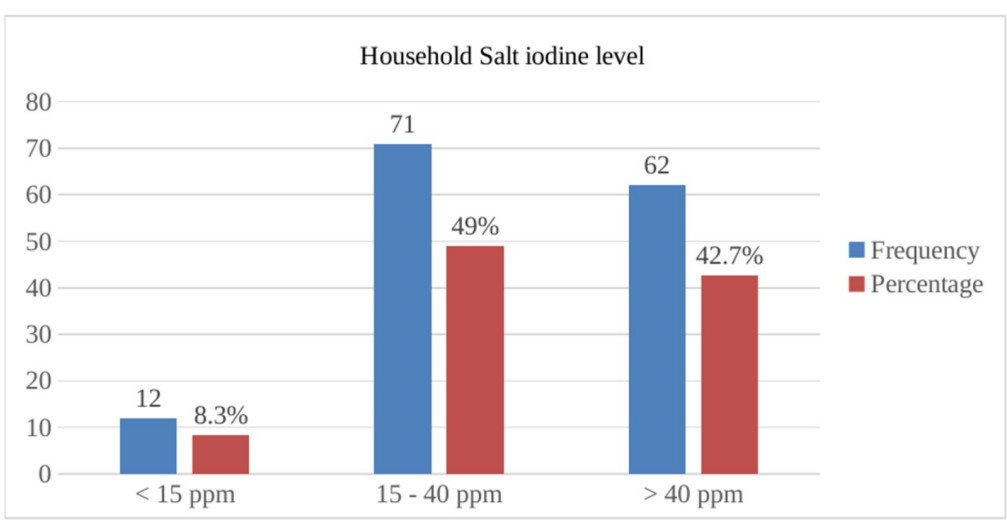

**Fig 1. Iodine content of household salt (ppm) consumed by pregnant women (N = 145) in BUNMAP cohort study, Butajira, Ethiopia.**

judged to have good knowledge. Nearly two third of the women had good knowledge with regard to iodized salt while one third of the study participants had poor knowledge (Table 2).

Most (91.2%) participants who bought non-iodized salt got it from the open market. The respondents tend to believe the salt obtained from the open market is non-iodized, while salt obtained from shops and supermarkets is iodized. Only 39(45.9%) of the 85 participants who claimed that they were using iodized salt knew that they can identify iodized salt by reading the label. The others claimed that they identify it by looking at its fineness (n = 20, 23.5%) or cleanliness (n = 26, 30.6%). The main reason cited for not buying iodized salt by the 60 respondents who thought their salt was not iodized was lack of knowledge about the benefit of iodine. Twenty-eight percent of women answered that iodized salt is more expensive than non-iodized salt and they opt to buy the less expensive one.

**Table 2. Knowledge & practice of pregnant women on iodized salt utilization (N = 145) in BUNMAP cohort study, Butajira, Ethiopia, February 2019.**

| Variables | Frequency (%) |
|---|---|
| Knowledge | |
| Good | 84(57.9) |
| Poor | 61(42.1) |
| Type of salt purchased for daily use (respondent's answer) | |
| Iodized | 85 (58.6) |
| Non-iodized | 60 (41.4) |
| Way of identification of iodized salt by respondent | |
| Being finer | 20 (23.5) |
| Being cleaner | 26 (30.6) |
| Reading label | 39 (45.9) |
| Reason for not buying iodized salt (n = 60) | |
| More expensive | 17 (28.3) |
| Not knowing the benefit | 34 (56.7) |
| Not available | 9 (15) |
| Kind of salt consumed | |
| Packed/ powdered | 65 (44.8) |
| Loose /coarse | 80 (55.2) |
| Salt purchasing frequency | |
| Weekly | 21 (14.5) |
| Every two weeks | 73 (50.3) |
| Monthly | 51 (35.2) |
| Salt purchasing place | |
| Open market | 70 (48.3) |
| Retail shop | 75 (51.7) |
| Salt storage place | |
| Dry & cool area | 119 (82.1) |
| Moisture & fire/sunlight exposed area | 26 (17.9) |
| Type of salt container | |
| with cover | 124 (85.5) |
| without cover | 21 (14.5) |
| Salt adding time | |
| At the beginning of the cooking process | 21 (14.5) |
| In the middle of the cooking process | 77 (53.1) |
| At the end of cooking & after cooling | 47 (32.4) |

### Iodine status of the pregnant women

**Quality control.** Six urine quality control samples with different iodine concentrations (27.1, 60.7, 84, 189.9, 282.3 and 523.6 ppb) were generously offered by the Lieberman Lab at the University of Notre Dame (USA), which is accredited for urinary iodine analysis by the Center for Disease Control's Ensuring the Quality of Urinary Iodine Procedures (EQUIP) program. The iodine concentration of blinded control samples was determined by the analyst and compared against the true value. The values obtained in the lab were 25.4, 73.2, 95.6, 167.2, 231.6 and 471.6 ppb respectively in the order mentioned above for the reference lab. All values obtained are in the acceptable range per the recommended guideline.

The study participants' urinary iodine levels are categorized according to the 2018 UNICEF standards, which include both a requirement that the median UIC for the population should fall in the range between 150 µg/L and 249 µg/L, and a requirement that no more than 20% of the population tested should show UIC below 50 µg/L. The data are shown in Fig 2. The urinary iodine content (UIC) levels measured among the study population ranged from 12.4 to 401.8 µg/L. The median UIC was 151.2 µg/L (IQR = 85.5–236.2 µg/L) with 0.69% of women (n = 1) showing less than 20 µg/L. Nearly half (50.35%) of the women had urinary iodine concentrations of at least 150 µg/L. Of these women, 40 (27.59%) are considered to have sufficient iodine intake as their urinary iodine level was between 150 µg/L and 249 µg/L. The remaining 33(22.76%) women had a UIC level between 250 µg/L and 450 µg/L. No woman had a UIC over 450 µg/L. The percentage of pregnant women with urinary iodine concentration of less than 50 µg/L is 9.7% (14/145), which is less than 20% of the cohort tested. Thus by the 2018 UNICEF guideline the study population cannot be judged as iodine deficient [18].

### Urinary iodine concentration and its associated factors

There was no significant association between the age group and UI level (*AOR* = 1.05; *CI* = 0.42–2.43) as shown in Table 3. The median UIC values for age groups 18–24, 25–34, and 35–49 were 149.85, 150.2, and 155.5 µg/L respectively.

The median UIC for women in their first trimester was 138.7µg/L, for the second trimester it was 170.7 µg/L, and for the third trimester it was 145.7 µg/L. Among the 23 (15.9%) pregnant

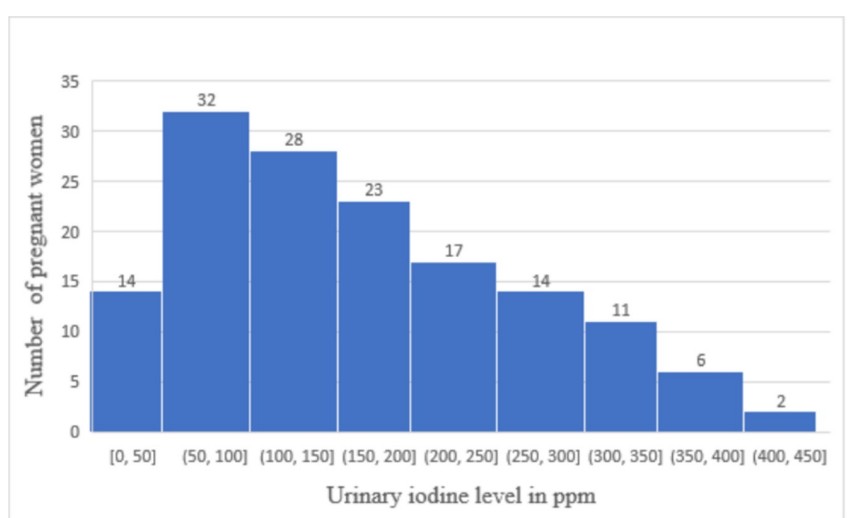

**Fig 2. Distribution of spot urine iodine concentration of pregnant women (N = 145) in BUNMAP cohort study, Butajira, Ethiopia.**

**Table 3. Median UIC of pregnant women (n = 145) according to age, trimester, salt iodine level, literacy level, and residence in BUNMAP cohort study, Butajira, Ethiopia, February, 2019.**

| Characteristics | N (%) | Median UIC(IQR), µg/L |
|---|---|---|
| Maternal age (years old) | | |
| 18–24 | 50(34.5) | 149.9 (79–235.65) |
| 25–34 | 85(58.6) | 150.2 (92.3–244.6) |
| 35–49 | 10(6.9) | 155 (108.7–214.4) |
| Trimester | | |
| First | 23(15.9) | 138.7(66.7–236.6) |
| Second | 47(32.4) | 170.7(96–235.8) |
| Third | 75(51.7) | 145.7(88.6–238.9) |
| Residence | | |
| Rural | 58(40) | 144.1(73.7–234.4) |
| Urban | 87(60) | 153.3(88.6–250.3) |
| Salt iodine level (ppm) | | |
| < 15 | 12(8.3) | 112.3(59.7–206.7) |
| 15–40 | 71(49) | 129.4(63.7–236.6) |
| > 40 | 62(42.7) | 182.9(125–252.1) |
| Educational status | | |
| No formal education | 40 | 140.5(68.5–237.1) |
| Have formal education | 105 | 153.3(88.5–236.2) |

women in their first trimester, 13 (56.5%) had less than 150 µg/L UIC and 10 (43.5%) had more than 150 µg/L UIC. Iodine deficiency was higher (56.5%) in the first trimester of pregnancy compared to the second (42.5%) and third (52%) trimesters.

Multivariable logistic regression analysis (MVLRA) of factors associated with Iodine sufficiency or deficiency had been calculated and described in Table 4. There was no significant association between age, trimester, residence & UIC of pregnant women. There was a significant association between having formal job (*AOR* = 2.56; *CI* = 1.11–5.96) and iodine sufficiency in pregnant women (Table 4).

## Discussion

The Ethiopian Standards Agency specifies that both iodized common and table salts should contain 36–80 ppm of iodine level [19]. The quality of iodized salt measured from the household salt samples provided by our BUNMAP participants is inconsistent (range = 0–95.3 ppm). However, the mean iodine level in the salt (35 ppm) in this study is greater than previous studies done in Northwest Ethiopia (20.79 (± 11.56) ppm) [20] and in Sidama zone, South Ethiopia [18.08 (±1.6) ppm] [21]. This higher iodine level shows improved processing, packaging, and storage of iodized salt in the country [2].

The amount of iodine in iodized salt can vary for different reasons. The higher iodine content observed in some samples may be due to the higher amount of iodine added during the iodization process or uneven distribution of iodine in the iodized salt. The lower content of iodine found in some salt samples may be due to inadequate iodization or loss of iodine through inappropriate packaging, storage, or distribution practices [2].

In this study, a major proportion (91.7%:95% *CI* = 87.2% - 96.2%) of households were consuming adequately iodized salt (≥ 15 ppm). Of these households, 62 (42.7%) provided salt samples that contained more than 40 ppm iodine. Only 12 (8.3%) of the household salt samples contained less than 15 ppm (Fig 1). According to the WHO standard, elimination of IDD

**Table 4. Multivariate logistic regression analysis on factors associated with iodine status of pregnant women (N = 145) in BUNMAP cohort study, Butajira, Ethiopia, February, 2019.**

| Characteristics | Iodine Status | | Crude OR (COR) | Adjusted OR |
|---|---|---|---|---|
| | <150 µg/L Deficient | ≥150 µg/L Sufficient | | |
| Age distribution (years) | | | | |
| 18–24 | 25 (17.2%) | 25 (17.2%) | 1.00 | 1.00 |
| 25–49 | 47 (32.4%) | 48 (33.1%) | 1.02 (0.52–2.03) | 1.05 (0.42–2.43) |
| Residence | | | | |
| Rural | 31 (53.4%) | 27 (46.6%) | 1.00 | 1.00 |
| Urban | 41 (47.1%) | 46 (52.9%) | 1.29 (0.66–2.51) | 1.09 (0.39–3.04) |
| Educational level | | | | |
| No formal education | 22 (55.5%) | 18 (44.5%) | 1.00 | 1.00 |
| Have formal education | 50 (47.6%) | 55 (52.4%) | 1.1 (0.75–1.61) | 0.89 (0.32–2.52) |
| Occupation | | | | |
| Housewife | 53 (56.4%) | 41 (43.6%) | 1.00 | 1.00 |
| Employed &Merchants | 19 (37.3%) | 32 (62.7%) | 2.18 (1.08–4.38) | 2.50 (1.12–5.67) |
| Family size | | | | |
| < 5 persons | 53 (51%) | 51 (49%) | 1.00 | 1.00 |
| ≥ 5 persons | 19 (46.3%) | 22 (53.7%) | 1.2 (0.58–2.48) | 1.4 (0.46–4.22) |
| Monthly income | | | | |
| ≤ 500 | 26 (56.5%) | 20 (43.5%) | 1.00 | 1.00 |
| 501–1000 | 17 (41.5%) | 24 (58.5%) | 1.84 (0.78–4.30) | 0.79 (0.35–1.76) |
| 1001–5000 | 29 (50%) | 29 (50%) | 1.30 (0.60–2.83) | 1.75 (0.74–4.11) |
| Gestational period | | | | |
| First trimester | 13 (56.5%) | 10 (43.5%) | 1.00 | 1.00 |
| Second trimester | 20 (42.5%) | 27 (57.5%) | 1.75 (0.64–4.8) | 1.54 (0.47–5.03) |
| Third trimester | 39 (52%) | 36 (48%) | 1.20 (0.47–3.07) | 1.44 (0.46–4.47) |
| Number of children | | | | |
| 0 | 21 (43.8%) | 27 (56.2%) | 1.00 | 1.00 |
| 1 | 25 (62.5%) | 15 (37.5%) | 0.47 (0.20–1.10) | 0.57 (0.13–2.57) |
| 2–4 | 22 (47.8%) | 24 (52.2%) | 0.85 (0.38–1.91) | 0.24 (0.05–1.07) |
| ≥ 5 | 4 (36.4%) | 7 (63.6%) | 1.36 (0.35–5.27) | 0.52 (0.13–2.12) |
| Salt iodine level | | | | |
| Not adequate | 10 (66.7%) | 5 (33.3%) | 1.00 | 1.00 |
| Adequate | 65 (48.9%) | 68 (51.1%) | 1.47 (0.44–4.85) | 1.24 (0.35–4.32) |
| Heard about IDD | | | | |
| Yes | 42 (50%) | 42 (50%) | 1.00 | 1.00 |
| No | 30 (49.2%) | 31 (50.8%) | 1.03 (0.5–1.99) | 1.40 (0.58–3.39) |
| Add salt while cooking | | | | |
| At the beginning | 10 (47.6%) | 11 (52.4%) | 1.00 | 1.00 |
| In the middle | 39 (50.6%) | 38 (49.4%) | 0.89 (0.34–2.33) | 0.76 (0.24–2.44) |
| At the end | 23 (48.9%) | 24 (51.1%) | 1.21 (0.42–3.47) | 1.30 (0.35–4.84) |
| Consumption of cabbage | | | | |
| Yes | 60 (48.4%) | 64 (51.6%) | 1.00 | 1.00 |
| No | 12 (57.1%) | 9 (42.9%) | 1.42 (0.56–3.62) | |
| Cabbage consumption per week | | | | |
| None | 12 (57.1%) | 9 (42.9%) | 1.00 | 1.00 |
| Once | 32 (54.2%) | 27 (45.8%) | 1.13 (0.41–3.07) | 0.64 (0.18–2.26) |
| Twice | 18 (41.9%) | 25 (58.1%) | 1.85 (0.64–5.32) | 0.85 (0.30–2.40) |
| More than twice | 10 (45.4%) | 12 (54.6%) | 1.60 (0.48–5.34) | 1.44 (0.48–4.30) |

should be achieved in this area since more than 90% of the households consume adequately iodized salt [2]. The observed percentage of adequately iodized household salt was in line with the 2016 report of the Ethiopian Demographic and Health Survey (EDHS), which found the coverage of adequately iodized household salt as 89% [22]. The coverage rate was significantly higher than previous studies done in Gondar town, Amhara region (2013, 28.9%) [23], Laelay Maychew district, Tigray region (2015, 33%) [24], Dabat district, Amhara region (2017, 33.2%) [25], Assosa town, Benishangul Gumuz region (2014, 26.1%) [26] and from a 2011 report of EDHS (2011, 15%) [27]. The improved availability of adequately iodized salt and the enhanced iodine level of salt in the current study show the impacts of more effective implementation of legal regulations in recent years than in the past, improved availability, and accessibility of iodized salt in the market, and enhanced health education on utilization of iodized salt [22].

All the respondents reported storing their salt for less than two months after purchase. The majority of the participants (82.1%) stored salt in a dry place (on the shelf or table) and more than three-quarters of participants (85.5%) stored salt in a tightly closable container. These practices of storing salt slow the loss of iodine from exposure to moisture, heat, and light [28]. The iodine levels of the salt samples in this study did not show any significant association with the storage place or the type of container in which the salt was kept.

It is recommended by WHO that iodized salt should be added to food after cooking to reduce the loss of iodine [29]. Cooking practices can impact the availability of the iodine in the cooked food. There was no practice of salt treatment before consumption in this community, in contrast with other areas where household salt may be washed and dried in direct sunlight or over afire before consumption [30]. A study in Haiti demonstrated that 5 seconds swirling coarse salt in a bowl of water leads to 60–70% retention of the iodate, while 40–30% of the iodate lost by the washing procedure [31]. Sixty percent of iodine can be lost during cooking. Nearly half (53.1%) of the participants usually added salt in the middle of the cooking process while 32.4% added salt at the end of cooking.

In this study, there was no association of iodine content with whether the salt was packed or loose, in contrast to earlier studies in Gondar, North West Ethiopia [20], Lalo Assabi District, West Ethiopia, [32] and Robe town, South Central Ethiopia [33]. Loss of iodine due to environmental factors from non-packaged salt was higher than from packaged salt in a previous study [29]. However, the duration of salt storage at home was much longer for the study conducted in Gondar and Robe; in this study, many respondents purchased salt frequently and storage time at their home was less than a month. The salt samples collected in this study were iodized using potassium iodate, rather than potassium iodide. As iodates are less soluble and more resistant to oxidation than iodide, the salt iodine content remains relatively constant under different environmental conditions (moisture, heat, and sunlight) even in loose salt [34–36]. Salts containing potassium iodate on heating retain a high percentage of their original iodine content while salts iodized with potassium iodide loose a considerable amount [37].

UIC determination by the Sandell-Kolthoff method is a challenging analytical task, but our quality control results showed adequate proficiency with the method. The difference between the laboratory-measured value and true value for a set of urine samples with known iodine content was in the acceptable range (Acceptable range = ± 30% of target value when target is less than 50 μg/L; ±25% when target value 50 to 100 μg/L; ±20% when target value >100 μg/L) [38].

The household salt iodine levels in this study should be associated with an adequate urinary iodine status of the population according to WHO criteria [2], and this was in fact observed. The observed median in the population of pregnant women in this study, 151 μg/L, was markedly higher than previous studies of population UIC conducted in 2013–2014 in Jimma

town, South West Ethiopia (48 μg/L) [16], Haramaya, South East Ethiopia (58.1 μg/L) [39] and Sidama zone (15 μg/L) [40]. The increase in median UIC may be caused by the documented higher availability of adequately iodized salt and its proper utilization in this area. However, the median UIC was lower than recent studies carried out in Nigeria (163.1 μg/L) [41], Spain (172 μg/L) [42], and Australia (189 μg/L) [43] and the median value was close to the lower range of the recommended level (150 μg/L). This population of pregnant women was only borderline iodine sufficient. Therefore, efforts should be strengthened to maintain median UIC in the recommended range in the future. It is also helpful to improve the UIC level by including iodine rich food such as dairy products in the diet.

No excessive iodine intake was found among the study participants. This area of Ethiopia has few bodies of water and there is little consumption of ocean fish & seaweed. Most excessive iodine intake is observed in coastal areas like Japan where consumption of ocean fish and seaweed is high [44].

Although spot urine tests cannot be used to assess iodine nutrition status of individuals, the heterogeneity of the UI values gives some insights about the severity of iodine deficiency in this population compared to other populations. Among the study participants, about half (49.65%) (CI: 41.6–57.7) had UI levels that would be categorized as iodine deficient. Among these, 57 (39.31%) of the levels were mildly deficient and 14 (9.65%) of the levels were moderately deficient. In similar studies conducted earlier in Ethiopia, the prevalence of UI levels corresponding to iodine deficiency was found to be 89.9% for pregnant women in Jimma, south western Ethiopia and of this 12.3% were severely iodine deficient [16]. In another study in Sidama, about 60% of the women had lower than 20 μg/L UIC, and only 10% had UIC levels higher than 50 μg/L [40]. 82.8% of iodine deficiency was discovered in Haramaya district [39]. 61.4% iodine deficiency was reported in Northwest Ethiopia among pregnant women [45]. Although the results we found in this study were improved over these earlier reports, the prevalence of UIC levels below 150 μg/L in this study was higher than the prevalence of UIC levels below 150 μg/L in other developing countries. For example, 42.5% prevalence of iodine deficiency was reported among pregnant women in study done in Ghana [46] & 63% of pregnant women had sufficient iodine nutrition in India [47].

Therefore, half of the women and their fetuses are observed to be vulnerable to the unwanted outcome of iodine deficiency in this study. Moderate iodine deficiency during pregnancy may cause maternal hypothyroidism, which is associated with gestational diabetes mellitus, gestational hypertension, severe preeclampsia, cesarean sections, and preterm births [48]. Mild iodine deficiency during pregnancy may also decrease child cognition [6]. Severe iodine deficiency among the women was nearly nil (0.7%) in this study. As the percentage of severe iodine deficiency was very small (< 20%), the women in this area are not vulnerable for outcomes caused by severe iodine deficiency such as increased risk of child loss (miscarriage, fetal or neonatal death) [49].

There was no significant association ($AOR$ = 1.54; $CI$ = 0.47–5.03) and ($AOR$ = 1.44; $CI$ = 0.46–4.47) between the trimester and iodine status of the women in this study. However, a study in Japan reported the median UI/Creatinine increased from 185.4 μg/g Cr in the first trimester to 258.9 μg/g Cr in the second trimester and decreased slightly to 237.9 μg/g Cr in the third trimester [5]. The urinary iodine was slightly lower in the first trimester of pregnancy than the second and third trimesters of pregnancy in Jimma, Ethiopia (indicating that at the first half of gestation, there is high need of iodine) [16]. A median UIC of 92 μg/L in first, 96 μg/L in the second and 91 μg/L in the third trimester was observed in another study in Norway [50]. There was no specific pattern of urinary iodine levels with parity. This is consistent with the finding that urinary iodine excretion does not correlate with parity or gravidity [51].

There was no statistically significant association between age, trimester, residence & UIC of pregnant women in this study like another study in Ghana [46].

Iodine deficiency was lower among pregnant women who live in urban areas relative to pregnant women living in the rural areas. This might be because urban residents had more access to information about benefit of iodine, IDD & proper utilization of iodized salt. However, no statistically significant association was observed between the residence of pregnant women and iodine deficiency (*AOR* = 1.09; *CI* = 0.39–3.04). Though there was no statistically significant association (*AOR* = 1.75; *CI* = 0.74–4.11) between monthly family income and iodine sufficiency of the pregnant women, iodine deficiency was more prevalent in pregnant women whose monthly income is less than or equal to five hundred Ethiopian birr. Lower monthly income could lead to poor nutrition in the family, resulting in deficiency of minerals & vitamins. A study on the Tasmanian population showed no association between socioeconomic status and UIC [52].

Women in business or government employment had higher urinary iodine concentration (better iodine status) than ones who were housewives (*AOR* = 2.50; *CI* = 1.12–5.67). Being employed improves income which in turn improves the access to iodine-rich foods. It may also improve access to information on importance of good diet during pregnancy. Other studies also show positive association between having formal occupation and UIE level [46, 47].

Two thirds (66.7%) of women who consumed inadequately iodized salt were iodine deficient, while half (48.9%) of women who consumed adequately iodized salt were iodine deficient. The observed positive association was expected because the UIC is affected by recent iodine intake [2]. But the association was not significant (*AOR* = 1.24; *CI* = 0.35–4.32), since spot urine samples are subject to large intra individual variation [43].

Loss of iodine during cooking by evaporation result in less amount of iodine gained from the food [53]. One-third of pregnant women add salt at the end of cooking, this will cause a positive effect on the UIE of the women as it minimizes the heat exposure and subsequent evaporation of iodine in the cooking process. Subsequently the women will get sufficient iodine amount from their final food ready for consumption.

It has long been known that certain other ions in the diet, especially those that are large and polarizable like thiocyanate (significant amounts are found in certain vegetables) and perchlorate (which may be present in water), act as competitive inhibitors to iodide; these are often termed goitrogens [54]. Cultivation of goitrogenic foods such as cassava in this area is not common, but cabbage is widely consumed. Though the consumption of cabbage is common, there was no significant association between the urinary iodine concentrations and cabbage consumption.

## Limitations

This study had used spot urine collection method for urinary iodine determination in pregnant women. Thus the UIC levels found may be prone to dilutions and concentrations in relation to the volume of water/fluid intake by the study participants. The study assessed only 145 pregnant women, thus the findings and conclusions may not extrapolate to all pregnant women in the study area or in the country as a whole.

## Conclusion

The median iodine level of the collected salt samples in this study was 34.9 ppm (IQR = 24.2–44.6). Nearly all household salt (91.7%) consumed by pregnant women in this area was adequately iodized. Most of the pregnant women had good knowledge and proper practice about iodized salt. The majority of the participants (82.1%) stored salt in a dry place and more than

three-quarters of participants (85.5%) stored salt in a container with a lid closed tightly. The median UIC in this study was 151.2 μg/L (IQR = 85.5–236.2) as analyzed from spot urine which indicates optimum median UIC in pregnancy. Nearly half of (50.35%) women had optimal iodine nutrition (UIC equal to or above 150 μg/L). However, given at the median UIC was at the bottom of the recommended range, there is still wide spread moderate to mild iodine deficiency among some pregnant women. There was no significant association between UIC and the age, trimester, and residence of pregnant women. There was a significant association between having a formal job ($AOR$ = 2.56; $CI$ = 1.11–5.96) and iodine sufficiency in pregnant women. For sustainable prevention of IDD, regular monitoring of quality of salt and strengthening public awareness on its efficient utilization at household level is needed. Iodization Programs and health literacy campaigns should target populations at more risk like pregnant women so IDDs can be prevented effectively in the population.

## Supporting information

**S1 File. Data collection format.** All other data generated in this study was available on the Open Science Frame work (OSF) in the link https://osf.io/c2e79.
(DOCX)

## Acknowledgments

The authors are also grateful to Dr. Madeline Eberle, currently at Hikma Pharmaceuticals, Columbus, Ohio, USA, and Prof. Getahun Merga (Andrews University, MI, USA) for their assistance in obtaining reagents needed for this study. ST also acknowledges Addis Ababa University's female scholarship program for an MSc study stipend offered to her.

## Declaration

Approval for the study was obtained from the Ethical Review committee of the School of Pharmacy, Addis Ababa University (AAU) dated November 30, 2018 referenced as ERB/SOP/43/11/2018 and IRB of the College of Health Sciences, AAU. Purpose of the study was explained to all study participants, and written informed consent obtained. The study was carried out in accordance with relevant guidelines and regulations.

## Author information

Email Addresses: Senait Tadesse (ST) (Senipharma2000@gmail.com); Department of Pharmaceutical Chemistry and Pharmacognosy, Addis Ababa University, Ethiopia; Ayenew Ashenef (AA) (ayenew.ashenef@aau.edu.et); Department of Pharmaceutical Chemistry and Pharmacognosy, Addis Ababa University, Ethiopia; Ariaya Hymete (AH) (ariaya.hymete@aau.edu.et); Department of Pharmaceutical Chemistry and Pharmacognosy, Addis Ababa University, Ethiopia; Marya Lieberman (ML) (mlieberm@nd.edu); Department of Chemistry & Biochemistry, University of Notre Dame, USA; Seifu Hagos Gebreyesus (SG) (seif_h23@yahoo.com); Department of Public Health, Addis Ababa University, Ethiopia.

## Author Contributions

**Conceptualization:** Ayenew Ashenef.

**Data curation:** Seifu Hagos Gebreyesus.

**Formal analysis:** Ariaya Hymete, Seifu Hagos Gebreyesus, Ayenew Ashenef.

**Funding acquisition:** Seifu Hagos Gebreyesus, Ayenew Ashenef.

**Investigation:** Senait Tadesse, Ayenew Ashenef.

**Methodology:** Marya Lieberman, Ayenew Ashenef.

**Supervision:** Ariaya Hymete, Marya Lieberman, Seifu Hagos Gebreyesus, Ayenew Ashenef.

**Writing – original draft:** Senait Tadesse.

**Writing – review & editing:** Senait Tadesse, Marya Lieberman, Seifu Hagos Gebreyesus, Ayenew Ashenef.

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
