## [Decision Letter · Decision Letter 0]

13 Jul 2022

PONE-D-21-39936Iodine status, household salt iodine content and Knowledge Attitude and Practice (KAP) assessment among Pregenant women in Butajira, South central EthiopiaPLOS ONE

Dear Dr. Ashenef,

Thank you for submitting your manuscript to PLOS ONE. After careful consideration, we feel that it has merit but does not fully meet PLOS ONE’s publication criteria as it currently stands. Therefore, we invite you to submit a revised version of the manuscript that addresses the points raised during the review process.

We look forward to receiving your revised manuscript.

Kind regards,

Bijaya Kumar Padhi, PhD, MPH

Academic Editor

PLOS ONE

Journal Requirements:

A clean copy of the edited manuscript (uploaded as the new *manuscript* file.]

“The authors thanks Addis Ababa University (AAU) graduate student research support programme.  AAU thematic research project   grant supported us for the Cohort study. The Ministry of Innovation and Technology, Government of Ethiopia National Innovation Award to AA helped us in some laboratory works. “

“The authors thanks Addis Ababa University (AAU) graduate student research support programme.  AAU thematic research project   grant supported us for the Cohort study. The Ministry of Innovation and Technology, Government of Ethiopia National Innovation Award to AA helped us in some laboratory works.”

Reviewers' comments:

Reviewer's Responses to Questions

**Comments to the Author**

1. Is the manuscript technically sound, and do the data support the conclusions?

Reviewer #1: Partly

Reviewer #2: Yes

2. Has the statistical analysis been performed appropriately and rigorously? 

Reviewer #1: Yes

Reviewer #2: Yes

3. Have the authors made all data underlying the findings in their manuscript fully available?

Reviewer #1: Yes

Reviewer #2: Yes

4. Is the manuscript presented in an intelligible fashion and written in standard English?

Reviewer #1: Yes

Reviewer #2: Yes

5. Review Comments to the Author

Reviewer #1: The manuscript reports carefully conducted research on pregnant women, a group for whom iodine adequacy is critical. The use of blinded/masked quality control samples adds confidence to the study results.

The paper is clearly written, but unfortunately the authors may not have been aware of a crucial report published by UNICEF in 2018 (based on the work of a Technical Working Group Meeting on Research Priorities and Determination of Population Iodine Status). One statement from that report is: “With currently available methods, the mUIC can only be used to define population iodine status and not to quantify the proportion of the population with iodine deficiency or iodine excess”. This report updates the WHO/UNICEF/ICCIDD Guide for Programme Managers mentioned in this manuscript.

The author and title for the updated report follow:

UNICEF. (2018) Guidance on the Monitoring of Salt Iodization Programmes and Determination of Population Iodine Status. 19 pp.

Please consider updating both text and tables to be consistent with newest recommendations from the Technical Working Group.

Editorial Comments:

Line 19. Reference 7 is cited to say that “around 2 billion people have insufficient iodine nutrition worldwide”. Progress has been remarkable for salt iodization globally, so the newest data should be located. A reference about iodine deficiency in 2007 needs to be updated.

Line 59. The BUNMAP is a ..

Line 96. 65.7% does not match Table 1.

Line 108. ..”only 74 out of 145”……..The “only" seems odd given that 74 out of 145 is more than half.

References #2 and #13 seem to be identical.

Reviewer #2: 1. Nowhere in the questionnaire, attitude related data was mentioned. Remove attitude part from the title

2. How did you arrive at the sample size of 152 ? Provide justification for that sample size.

3. Justify the reason for collecting spot urine sample, but not 24 hour urine sample

4. Assessed the salt the participants are using, how did you ensure that this is the only brand of salt they are generally using?

5. Multivariable logistic regression analysis (MVRLA) tells about association, not correlation. Please correct in the manuscript

6. Some typos are there in the article. It was mentioned as multiple regressions in the results part, where it should be mentioned as Multivariable logistic regression analysis

7. Table 4 – Where is crude OR?

8. Why did you include all the variables in MVLRA when the conventional practice is to include only those variables with p-value <0.2 in BVLRA?

9. Where is attitude related data?

10. Table 1, V1, Knowledge as Good and Poor. Justify the operational definition of answering 2 out of 4 questions on knowledge correctly as good? Why only 2 grades? Why cant be spectrum of grades from Excellent – Very poor, justify?

11. “The average iodine level of household salt in this area was in the recommended range, and only 74 out of 145 samples were outside the recommended range.” Its not only 74, it is a huge chunk (> 50%). Please acknowledge that >50% are consuming salt whose Iodine PPI is out of the range

12. This study assessed only 145 women out of the 832 cohort. So, study findings might not be extrapolated to all the Pregnant woman of Butajira. Please comment on this external validity issue in Discussion/ limitations.

6. PLOS authors have the option to publish the peer review history of their article (what does this mean?). If published, this will include your full peer review and any attached files.

Reviewer #1: No

Reviewer #2: **Yes: **Dr. P. Siva Santosh Kumar

---

## [Author Response · Author response to Decision Letter 0]

3 Sep 2022

Rebuttal Letter

August 27, 2022.

We want to sincerely thank you and appreciate the editor and the two reviewers’ on the comments on our manuscript titled 

“Iodine status, household salt iodine content, Knowledge and Practice assessment among pregnant women in Butajira, South central Ethiopia PONE-D-21-39936”.

The comments are very valuable and help us to significantly improve our manuscript.

Below is our response comment by comment.

AyenewAshenef, 

On behalf of the authors.

Academic Editor

 Publishing and depositing the laboratory protocol:

Answer: As the laboratory methods used in this study are widely used standardized ones and the methods being described in the manuscript hence publishing as protocol in PLOS is not applicable in this manuscript.

 Ensuring the PLOS style requirements in the manuscript. With respect to formatting and other issues.

Answer: The manuscript had been revised keenly considering the PLOS One requirements.

 Inclusion of tables in the body of the manuscript at appropriate places and Supplementary information

Answer: Tables had been included in the manuscript rather than at the end. Supplementary information is also included which is our study tool.

 Copy editing thoroughly

Answer: Our manuscript had been copy edited thoroughly. Prof. Marya Lieberman one of the authors is a native English Speaker from USA. She had worked thoughtfully on the language issues as well as on the scientific content of this manuscript to meet the expectations of this journal.

 Stating funding related text on the acknowledgement

Answer: We had deleted that section to comply with the journal. We had stated funders in the funding section and other helps in the acknowledgement section.

6. Please include a separate caption for each figure in your manuscript.

Answer: A separate caption for each figure had been included

 Response to the reviewer’s 

Reviewer #1: The manuscript reports carefully conducted research on pregnant women, a group for whom iodine adequacy is critical. The use of blinded/masked quality control samples adds confidence to the study results. The paper is clearly written, but unfortunately the authors may not have been aware of a crucial report published by UNICEF in 2018 (based on the work of a Technical Working Group Meeting on Research Priorities and Determination of Population Iodine Status). One statement from that report is: “With currently available methods, the mUIC can only be used to define population iodine status and not to quantify the proportion of the population with iodine deficiency or iodine excess”. This report updates the WHO/UNICEF/ICCIDD Guide for Programme Managers mentioned in this manuscript.

The author and title for the updated report follow:

UNICEF. (2018) Guidance on the Monitoring of Salt Iodization Programmes and Determination of Population Iodine Status.19 pp.

Please consider updating both text and tables to be consistent with newest recommendations from the Technical Working Group.

Answer: Thank you for bringing this important guideline to our attention. Currently the manuscript had been revised based on this statement and this technical guideline. All parts including the abstract, body and conclusion had been re-phrased by taking this fact in to consideration in our revision.

Editorial Comments:

Line 19. Reference 7 is cited to say that “around 2 billion people have insufficient iodine nutrition worldwide”. Progress has been remarkable for salt iodization globally, so the newest data should be located. A reference about iodine deficiency in 2007 needs to be updated.

Answer: Again thanks for this critical comment and we had updated this fact and included recent progresses around the globe on this issue (Lines 19-27).

Line 59. The BUNMAP is a ..

Answer: Corrected as such (line 80).

Line 96. 65.7% does not match Table 1.

Answer: Corrected with the proper value (64.8%) in the table (line 120).

Line 108. ..”only 74 out of 145”……..The “only" seems odd given that 74 out of 145 is more than half.

Answer: Corrected per the comment and the data (line 134)

References #2 and #13 seem to be identical.

Answer: These references also the same trend, their publication year is different in 2011 and 2016, thus mentioned as different references.

Reviewer #2: 1. Nowhere in the questionnaire, attitude related data was mentioned. Remove attitude part from the title

Answer: Attitude is removed from the title and other parts of the manuscript based on your comment.

2. How did you arrive at the sample size of 152 ? Provide justification for that sample size.

 Answer: This is based on the sample size calculation of the formula.

The sample size was calculated using a single population proportion formula (see below) based on the following assumption, expected prevalence of iodine deficiency 90 % (16), absolute sampling error of five, and a 95% confidence interval (17). Hence, the sample size was 152 with a10% non-response rate. The number of pregnant women enrolled in the cohort at the time of this data collection was 832. Hence, we applied a simple random sampling method to select women from enrolled pregnant women. 

N=(Z^2×P(1-P))/d^2 = (〖1.96〗^(2 )×0.9(1-0.9))/〖0.5〗^2 =138

Where

N= Sample size

P= Prevalence of iodine deficiency in pregnant women (16)

Z=Z-score 

d =Sampling error 

This had been included in the manuscript (line 68-79).

3. Justify the reason for collecting spot urine sample, but not 24 hour urine sample

Answer: The study population is mainly a subsistence farming illiterate community with health literacy rate very low. Hence adherence to 24 hour urinary iodine collection was impossible and non- manageable. It is mainly a subsistence farming community that had to perform daily routines by moving in the village for agricultural duties. It is also culturally unacceptable to hold/carry a container for urine collection for 24 hours. Nevertheless also 24 hrs collection gives best; spot urine determination is used in the scientific community for such studies.

4. Assessed the salt the participants are using, how did you ensure that this is the only brand of salt they are generally using?

Answer: Again the community is mainly a subsistence farming community. It is not usual to possess two or more brands of salt for their consumption. Nevertheless they were asked to bring their all available salt. However we did not experience a family that possesses more than one type/brand of salt in their household.

5. Multivariable logistic regression analysis (MVRLA) tells about association, not correlation. Please correct in the manuscript

Answer: Manuscript had been revised considering this comment that is accepted. However whenever other references that studied correlation were mentioned this term was used.

6. Some typos are there in the article. It was mentioned as multiple regressions in the results part, where it should be mentioned as Multivariable logistic regression analysis

Answer: We did our best to address such issues in the revision.

7. Table 4 – Where is crude OR?

Answer: Crude OR had been included in the revised manuscript (see table 4)

8. Why did you include all the variables in MVLRA when the conventional practice is to include only those variables with p-value <0.2 in BVLRA?

Answer: Although your comment is valid, including and showing all data helps for the reader to comprehend details. Based on such belief, we had included all. In fact such values were obtained only in the factors that inform the number of children in a family and Cabbage consumption once per week.

9. Where is attitude related data?

Answer: Attitude related data was not as such solidly studied and presented in the study thus the title attitude is removed. The same was done on all the contents of the manuscript too.

10. Table 1, V1, Knowledge as Good and Poor. Justify the operational definition of answering 2 out of 4 questions on knowledge correctly as good? Why only 2 grades? Why can’t be spectrum of grades from Excellent – Very poor, justify?

Answer: We inferred that in such community where health literacy is very low answering half of the four questions can be judged (it is the mean) as good while values less than mean as poor. Including many spectrums in four questions did not seem logical. If we include many more questions the study tool will be too bulky. 

11. “The average iodine level of household salt in this area was in the recommended range, and only 74 out of 145 samples were outside the recommended range.” It’s not only 74; it is a huge chunk (> 50%). Please acknowledge that >50% are consuming salt whose Iodine PPI is out of the range

Answer: This interpretation had been included in the revision.

12. This study assessed only 145 women out of the 832 cohort. So, study findings might not be extrapolated to all the pregnant woman of Butajira. Please comment on this external validity issue in Discussion/ limitations.

Answer: This limitation had been included in the manuscript in the limitations section/part even-though sample size is statistically calculated (line 359-365).

---

## [Editor Report · Decision Letter 1]

24 Oct 2022

Iodine status, household salt iodine content, knowledge and practice assessment among pregnant women in Butajira, south central Ethiopia

PONE-D-21-39936R1

Dear Dr. Ashenef,

We’re pleased to inform you that your manuscript has been judged scientifically suitable for publication and will be formally accepted for publication once it meets all outstanding technical requirements.

Kind regards,

Bijaya Kumar Padhi, PhD, MPH

Academic Editor

PLOS ONE